# Modeling the consequences of the dikaryotic life cycle of mushroom-forming fungi on genomic conflict

**Benjamin Auxier[1], Tamás L Czárán[2,3], Duur K Aanen[1]***

[1]Laboratory of Genetics, Wageningen University, Wageningen, Netherlands; [2]ELKH Centre for Ecological Research, Institue of Evolution, Budapest, Hungary; [3]MTA-ELTE Theoretical Biology and Evolutionary Ecology Research Group, Budapest, Hungary

**Abstract** Generally, sexual organisms contain two haploid genomes, one from each parent, united in a single diploid nucleus of the zygote which links their fate during growth. A fascinating exception to this is Basidiomycete fungi, where the two haploid genomes remain separate in a dikaryon, retaining the option to fertilize subsequent monokaryons encountered. How the ensuing nuclear competition influences the balance of selection within and between individuals is largely unexplored. We test the consequences of the dikaryotic life cycle for mating success and mycelium-level fitness components. We assume a trade-off between mating fitness at the level of the haploid nucleus and fitness of the fungal mycelium. We show that the maintenance of fertilization potential by dikaryons leads to a higher proportion of fertilized monokaryons, but that the ensuing intra-dikaryon selection for increased nuclear mating fitness leads to reduced mycelium fitness relative to a diploid life cycle. However, this fitness reduction is lower compared to a hypothetical life cycle where dikaryons can also exchange nuclei. Prohibition of fusion between dikaryons therefore reduces the level of nuclear parasitism. The number of loci influencing fitness is an important determinant of the degree to which average mycelium-level fitness is reduced. The results of this study crucially hinge upon a trade-off between nucleus and mycelium-level fitness. We discuss the evidence for this assumption and the implications of an alternative that there is a positive relationship between nucleus and mycelium-level fitness.

**\*For correspondence:**
duur.aanen@wur.nl

**Competing interest:** The authors declare that no competing interests exist.

## Editor's evaluation

Unions between equal partners can be destabilized by matings with third parties. In this paper the authors demonstrated that in fungi, 'stable unions' of two nuclei (dikaryons) are predicted to experience costs to vegetative fitness from investment in such mating opportunities. 'Open unions', in which third parties have access to the resources of established partnerships, are evolutionarily highly unstable. This paper will be of general interest to those who study evolutionary conflicts and to fungal geneticists.

## Introduction

Kin-selection theory shows that altruistic interactions between individuals, or between other replicating entities, can evolve between genetically related vehicles (*Hamilton, 1964*). Examples are altruistic interactions between individuals in social insects, cooperation between clonally related mitochondria within eukaryotic cells, and between clonal cells of multicellular individuals (*Bourke, 2011*; *Queller, 2000*). In sharp contrast to social altruistic interactions, sexual interactions often are avoided among genetically related and promoted among genetically unrelated individuals, either via

behavior in animals or via incompatibility systems in plants and fungi (*Herron and Freeman, 2014*). This difference in preferred relatedness creates a tension between sexual and social interactions. This is illustrated in parental conflict over resource allocation to offspring. In species where females can carry embryos from multiple males, paternal genomes compete to extract nutrients from a common uterus/ovary (*Haig, 1993*). The competition between genetic elements, in this case paternal alleles from different embryos, inevitably leads to conflict with the mother over resource provisioning. Any paternal competitive trait expressed in the embryo provides the selective ground for modifiers of maternal origin to repress these competitive traits (*Rice and Holland, 1997*). This parent-of-origin conflict over maternal resource provisioning can be suppressed by mechanisms such as genomic imprinting.

Most sexually reproducing organisms have mechanisms to police competition among the unrelated genetic elements or to re-establish high genetic relatedness upon mating. For example, social insects typically are monogamous, meaning that social interactions occur among full siblings (*Boomsma, 2009*). Clonal relatedness among organelles is maintained by uniparental transmission and a bottleneck during sexual reproduction (*Cosmides and Tooby, 1981*). Also, multicellular organisms typically develop by clonal outgrowth of a fertilized egg, and thus the cells are genetically identical barring de novo mutations (*Buss, 1987*). In diploids tension is reduced between the two gametic genomes that form the zygote, because cell fusion (plasmogamy) is immediately followed by nuclear fusion (karyogamy), aligning the fates of the two genomes until the next meiosis. Tension is reduced between the unrelated haploid genomes because the benefits of a competitive allele from one genome are shared equally with the other genome. However, Basidiomycetes, the fungal clade containing mushrooms, rusts, and smuts, provide an exception to this general principle.

In these fungi, the fusion of plasma membranes occurs between two multicellular gametes (monokaryons), but nuclear fusion is delayed until immediately prior to meiosis. The mated individual with two separated haploid nuclei, termed a dikaryon, is genetically similar to a diploid by having two copies of the nuclear genome per cell, although gene regulation may differ due to the compartmented genomes (*Banuett, 2015*; *Schuurs et al., 1998*). The dikaryotic state differs from diploidy as the separate haploid nuclei retain the ability to fertilize further monokaryons (reviewed in *Raper, 1966*). A consequence of this delayed karyogamy in Basidiomycetes is an increase in mating opportunities (*Anderson and Kohn, 2007*). Since the fertilization of a monokaryon is not associated with resource investment in the resulting new dikaryon, the dikaryotic life cycle corresponds to the retention of the male role (*Aanen et al., 2004*). This unique form of mating between a dikaryon and monokaryon is known as di-mon mating, often called Buller mating (*Buller, 1934*; *Quintanilha, 1938*). Di-mon mating, clearly impossible for a diploid organism, has been thought to be the main benefit of the dikaryotic lifestyle (*Raper, 1966* p.264). The exchange of nuclei between dikaryons is thought to be prevented in nature since fusion of two dikaryons triggers nonself recognition followed by apoptosis, quickly killing the fused cell (*Auxier et al., 2021b*; *Aylmore and Todd, 1986*). The prolonged mating opportunities of the dikaryon allow a nuclear genotype increased access to resources, as it retains its share of resources in the original dikaryon, and gains access to additional resources of the newly fertilized dikaryon. However, the benefits to a nucleus of this delayed karyogamy may carry a cost due to persistent tension in social interactions between the unrelated nuclei.

These di-mon matings present a competitive arena for the paired nuclei. Due to the high number of mating types in Basidiomycetes, both nuclei of a dikaryon are generally capable of fertilizing an unrelated monokaryon. This results in competition between the nuclei, the dynamics of which are poorly understood. Initially both nuclei appear to invade hyphae of the monokaryon, but ultimately only one of the two dikaryotic nuclear haplotypes fertilizes the entire monokaryon (*Anderson and Kohn, 2007*). Competitive success within the dikaryon could be based on variation in rates of nuclear division, migration, or percentage of hyphae colonized (*Anderson and Kohn, 2007*; *Raper, 1966*; *Vreeburg et al., 2016*). Evidence of variation for mating success has been found in natural systems, where replicate di-mon matings consistently result in the same fertilizing nucleus (*Ellingboe and Raper, 1962*; *Nieuwenhuis et al., 2011*; *Nogami et al., 2002*). In one study, the direct interaction between competing nuclei was found to determine success in a Buller mating, with a hierarchy of mating competitiveness (*Nogami et al., 2002*). However, there is also evidence that the interaction of the different nuclei with the receiving monokaryon determines mating success in a Buller mating (*Nieuwenhuis et al., 2011*). Whether this intraspecific variation in mating success will affect a population

depends on the frequency of di-mon matings as well as simultaneous mon-mon matings. However, since mating success is difficult to quantify, the magnitude of this variation remains unknown. Alternatively, variation in nuclear competitiveness could be maintained due to trade-offs with mycelium-level fitness components.

Two common measures of fungal fitness are somatic (mycelial) growth and spore production (*Pringle and Taylor, 2002*). While these undoubtedly capture only a fraction of actual fitness variation between individuals, fitness components related to aspects like enzymatic variation have not yet been described (*Allison et al., 2018*). For these sessile organisms, vegetative growth is required to explore nearby resources, and spore production is needed for dispersal to new sites. It has been thought that fungal vegetative growth and spore production may be under a life history trade-off (*Schoustra and Punzalan, 2012*; but see *Anderson et al., 2019*). Such a trade-off assumes that high vegetative growth rate to explore resources comes at the expense of either rapid or abundant spore production (*Gilchrist et al., 2006*; *Pringle and Taylor, 2002*).

Here we explore the consequences of di-mon matings in the dikaryotic life cycle for the proportion of fertilized monokaryons and for mycelium-level fitness components, assuming a three-way trade-off between growth, spore production, and mating success. We simulated the outcome for three life cycles: diploid, standard dikaryon (with di-mon matings) with retention of the male role, and a hypothetical open dikaryon with retention of the male and female role (similar to a life cycle without nonself recognition between dikaryons). While this open dikaryon is not known from nature, we explore potential consequences of such a dikaryon for the balance between selection among nuclei within mycelia and among mycelia. The unique life cycle of the standard dikaryon allows for additional matings through di-mon matings, but may come with the cost of selection at the level of the nucleus (*James, 2015*). Our results show that the frequency of unmated monokaryons is reduced in the standard dikaryon life cycle, but selection at the level of the nucleus reduces average dikaryon mycelium-level fitness. However, this cost is greatly reduced compared to a hypothetical life cycle where exchange between dikaryons is also allowed. Our simulations also show that these costs are increased if the loci associated with mating fitness are spread throughout the genome.

## Materials and methods
### The model
#### The arena
The 'substrate' on which the simulated mycelia live is two-dimensional, represented by a square lattice each site supporting a single mycelium, or section thereof. The lattice takes a toroidal topology with the first and the last site of each row/column being neighbors, so that the lattice has no edges. At time 0, the lattice is inoculated with a random initial pattern of spores from a pool of $m$ different mating types. Each inoculated site contains the monokaryotic mycelium sprouted from a single spore.

#### Nuclear fitness
The nuclear fitness of each spore (and the corresponding monokaryon and dikaryon) consists of three heritable (genetically encoded) components:

1. vegetative fitness $w_v$ which determines the rate $G$ at which the mycelium expands to neighboring empty sites: $G = g \cdot w_v$, where $g$ is the basic rate of mycelial growth.
2. reproductive fitness $w_r$ which determines the rate $R$ at which the mycelium produces spores: $R = r \cdot w_r$, where $r$ is the basic rate of spore production; note that only dikaryotic mycelium produces spores, so this fitness has an effective value of zero as a monokaryon.
3. mating fitness $w_m$ that determines the propensity of the nucleus to become part of a dikaryon upon mating.

These three fitness components are traded off so that any one of them can increase only at the expense of the other two in a linear fashion: $w_v + w_r + w_m = 1.0$. (*Grafen, 2014*).

#### Nonlinear fitness
We also explored the effects of nonlinear trade-offs on possible combinations of $w_v$, $w_r$, and $w_m$ such that any actual combination remains below the trade-off surface defined by $1 = (w_v^{\beta} + w_r^{\beta} + w_m^{\beta})$.

Results of the nonlinear fitness are shown in *Figure 4—figure supplement 1*. Unless otherwise stated, in our modeling $\beta$ = 1.

## Dikaryotic fitness

The vegetative and the reproductive fitness components ($w_v$ and $w_r$) of dikaryons depend on the nuclear fitness components ($w_v$ and $w_r$) of the two nuclei it harbors and on the dominance interaction, $\Theta$. For example, dikaryotic mycelia grow into neighboring sites at a rate $W_v = g \left[ w_{v,max} \cdot Q + w_{v,min} \cdot (1 - Q) \right]$, where $g$ is the basic mycelial growth rate, $w_{v,max}$ is the vegetative fitness of the fitter nucleus, and $\Theta$ specifies its phenotypic dominance over the less fit nucleus (of nuclear vegetative fitness $w_{v,min}$). $\Theta$ = 1.0 is the absolute dominance, $\Theta$ = 0.0 means absolute recessivity. Reproductive fitness (i.e. the spore production rate) of the dikaryon is calculated similarly.

## Mating

In all scenarios, monokaryotic mycelia of different mating types coming into spatial contact (i.e. those on neighboring sites) will sexually fuse and produce dikaryons if they are of different mating types. In case of more possible mates, the focal ('female') monokaryon chooses one of the compatible nuclei from the neighboring mycelia, with the chance that a given 'male' nucleus is chosen depending on its mating fitness $w_m$. The three mating scenarios (diploid: mon-mon only; standard dikaryon: mon-mon and di-mon; open dikaryon: mon-mon, di-mon, and di-di) differ in the number of nuclei competing for becoming one of the two actual nuclei taking over the fused mycelia. In mon-mon matings only the nuclei from monokaryons surrounding a focal receiving monokaryon; in di-mon matings the two nuclei of each dikaryon compete as well as surrounding monokaryons, and in the di-di scenario two of the four nuclei win, with the chances depending on $w_m$. The winning pair of nuclei spreads all over the spatially connected parts of the affected male and female mycelia, transforming them into the same dikaryon.

## Mutation

Spore production occurs with the sexual fusion of the two nuclei of the dikaryon, preceded by mutations during the dikaryotic state, and followed by meiosis. Mutations affect the nuclear fitness components ($w_v$, $w_r$, and $w_m$), such that mutant fitness ($w_v$, $w_r$, and $w_m$) is drawn from Gaussian ditributions with standard deviation $\sigma$ centered on the parental values and scaled back to satisfy $w_v + w_r + w_m$ = 1.0. The mutation step is shown in *Figure 1A* as a set of open circles denoting the starting fitness as a position on the ternary diagram, and the arrow showing the mutational step to the new fitness values.

## Number of fitness related loci

To partially (implicitly) simulate the genetics underlying the phenotypes of this model, we implemented a parameter, $\lambda$, affecting the inheritance of fitness phenotypes, ranging between 0 and 1. $\lambda$ = 0 corresponds to multiple loci throughout the genome, where recombination between the parents leads to offspring with the average phenotype of two parental nuclei. Alternatively, $\lambda$ = 1 indicates that fitness variance is located inside a single Mendelian locus, and offspring resemble either one parent or the other.

## Updating algorithm

One generation of the simulations consists of $N$ elementary updating steps; $N$ is the number of sites in the lattice. An updating step starts with choosing a random site, which if occupied dies with probability $d$. If the site is instead empty, the mycelia on the neighboring eight sites compete to occupy it, chances to win depending on their vegetative fitness. If the focal site is occupied and survives, it may engage in mating in the 'female' role, provided that it is capable of mating with at least one of its neighbors. Mating type, mating scenario compatibility, and mating success $w_m$ of the potential 'male' partners determine the actual outcome of mating, resulting in a dikaryon in accordance with the rules detailed in the *Mating* section above. If the focal site contains a dikaryotic mycelium, it may produce spores which are locally dispersed onto sites within the dispersal radius $R$ of the parental dikaryon, which germinate to monokaryotic mycelia in the next generation. One spore per site may survive on each empty site that has received spores; the survivor is chosen at random. Identical nucleotypes instantly fuse somatically upon spatial contact.

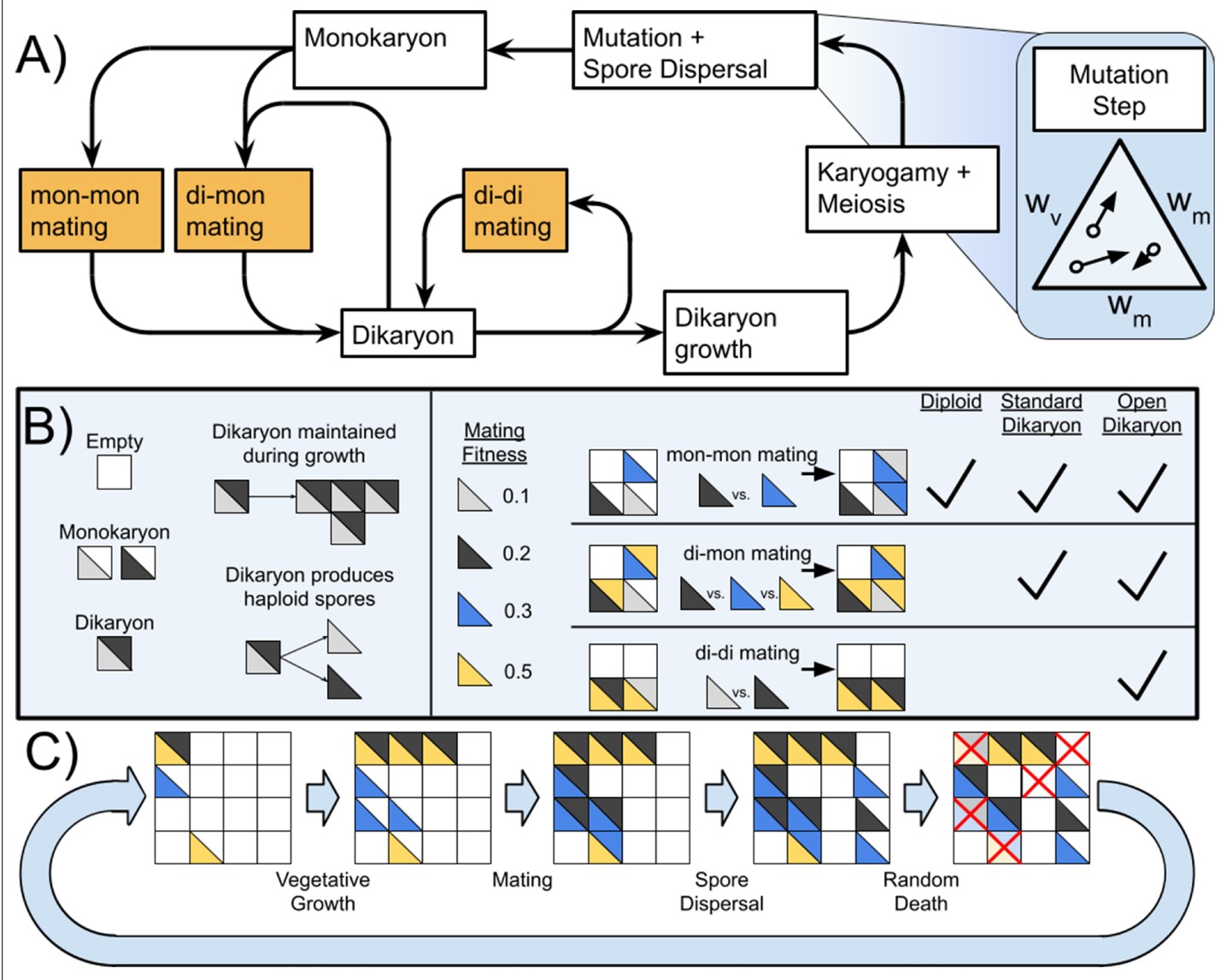

**Figure 1.** Life cycle of Basidiomycetes and schematic of model. (**A**) Life cycle of dikaryon showing the three different mating possibilities (mon-mon, di-mon, and di-di), as well as the separation between mating and karyogamy. (**B**) Shows some rules of the algorithm and three types of mating, and which matings are found with each of the three life cycles. (**C**) Example of the actions taken during one generation of the simulation. Note that in this diagram matings are shown as deterministic based on mating fitness, but in the simulations the outcome of competition is probabilistic.

The online version of this article includes the following figure supplement(s) for figure 1:

**Figure supplement 1.** Overview of simulation outcomes using higher mutation width of 0.05 instead of 0.01 used in other simulations.

**Figure supplement 2.** Overview of simulation outcomes displayed as in *Figure 2* but using higher basal growth rate of 0.5 instead of 0.1 used in other simulations.

The rules and the algorithm of the simulations are summarized in *Figure 1*. Pseudocode for the simulation can be found in Source Code File 1. Animations of sample runs for the Diploid, Standard, and Open Dikaryon can be found in Animation 1, Animation 2, and Animation 3, respectively.

## Competitions

To compare the competitive advantage of dikaryotic male function (DMF) (the standard dikaryon scenario), the simulation was modified to represent an allele conferring dikaryotic male function linked to the mating locus, having 15 mating types with this allele, and 15 without. Homokaryons of either type were reproductively compatible, and the case of heterozygotes was evalued for the DMF

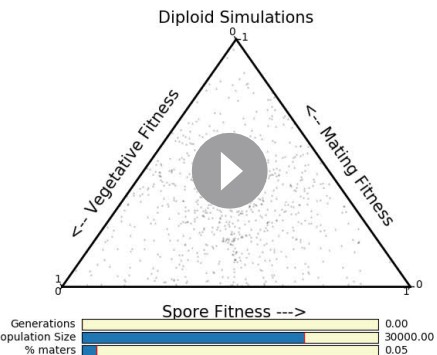

**Animation 1.** Animation of the first 100 generations of the diploid life cycle under standard conditions. Dots indicate individual nuclei. Bars indicate population size, and '% maters' which is the proportion of nuclei with >66% of fitness allocated to mating fitness, as in Figure 2.

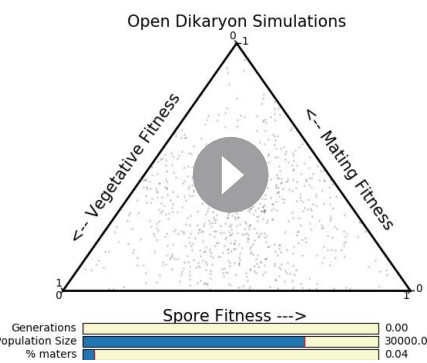

**Animation 3.** Animation as found in Animation File 2, except this time with open dikaryon life cycle.

allele being recessive, dominant, or co-dominant where only the nucleus with the DMF allele could participate in di-mon matings. Due to algorithmic complexity, we did not attempt a similar competition for dikaryotic female function (the open dikaryon scenario).

## Parameter settings

Due to the number of parameters, we could not assess all possible combinations. Preliminary results showed that simulations with grid sizes of less than 150 squares had stochastic outcomes (data not shown). To combat this, results shown here were performed with a grid size of 300 × 300, except data from Figures 5 and 6 where due to computational resources, smaller grid sizes of 200 × 200 were used instead. Except where otherwise specified the base parameters were: g = 0.1; $r$ = 1; d = 0.3; $\sigma$ = 0.01; grid sizes of 300 × 300 and simulations were run for 1000 iterations.

# Results

## Genetic assumptions have a strong influence on equilibrium fitness distributions

Initial simulations run under different scenarios (one locus/many loci, recessive/co-dominant/dominant) showed a strong influence of these parameters on the resulting nuclear fitness (*Figure 2*).

## Diploid

The diploid life cycle, where matings only occur between monokaryons, resulted in stable populations over the entire tested parameter space. When fitness is polygenic, the resulting nuclei cluster near the middle of the fitness space, without maximizing any specific trait. If instead fitness is monogenic the nuclei form a cluster with maximum spore production. When the mycelium-level phenotypes ($w_v$ and $w_r$) are fully dominant ($\Theta$ = 1.0) with monogenic fitness traits (rightmost column) the simulations result in one set of nuclei optimizing spore production and a much smaller cluster balancing vegetative growth and mating fitness, particularly with high growth rates. There is no obvious difference between the diploid simulations under the high spore (*Figure 2A*) or low spore (*Figure 2B*) state.

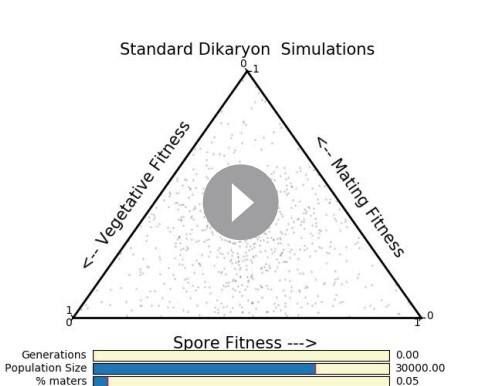

**Animation 2.** Animation as found in Animation File 2, except of the standard dikaryon life cycle.

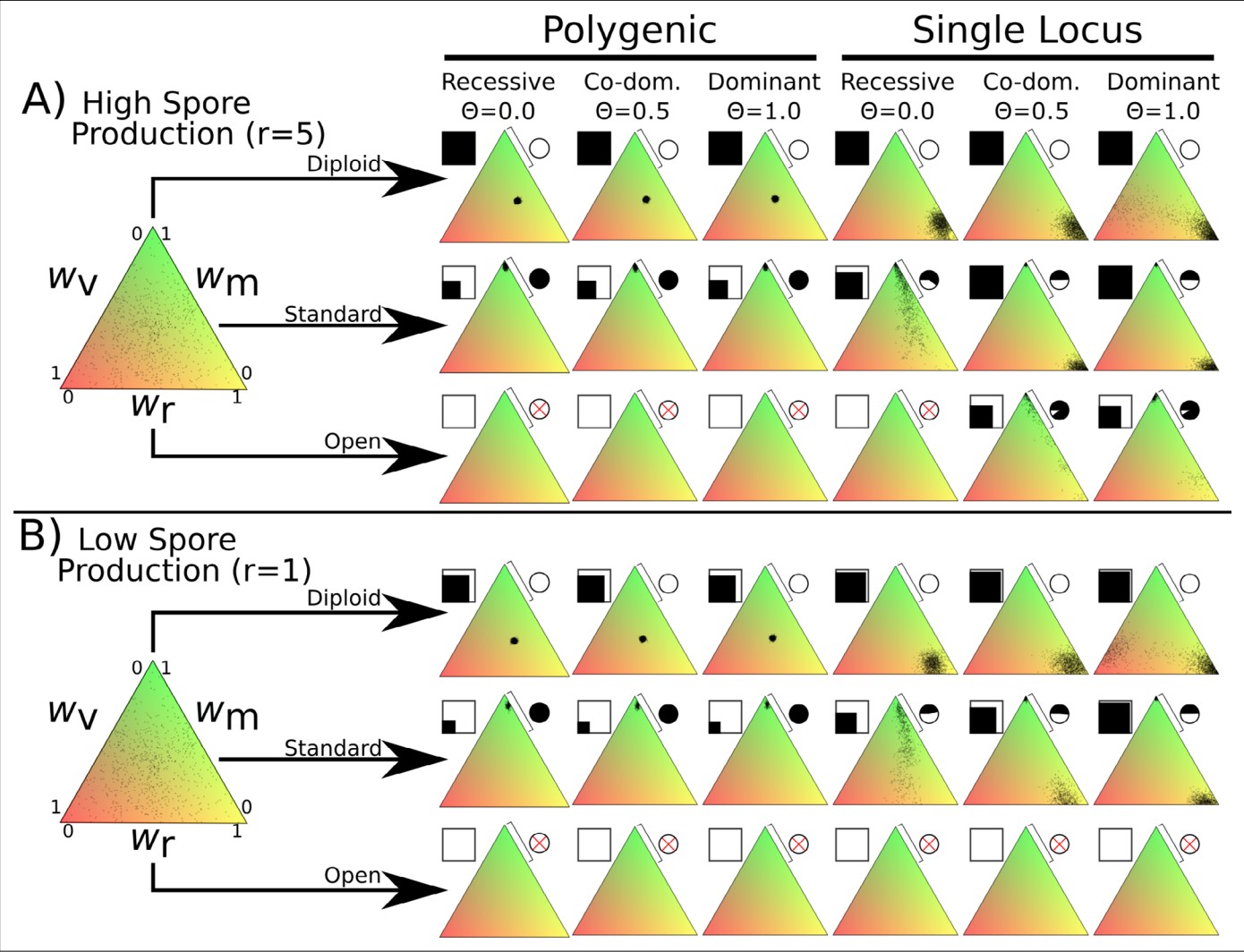

**Figure 2.** Fitness distributions after 1000 generations under different values of $\lambda$ and dominance coefficients for the three dikaryon scenarios with (**A**) high spore production ($r = 5$) or (**B**) low spore production ($r = 1$). Simulations were initialized with 60,000 homokaryotic individuals with initial fitness values drawn from a normal distribution, as shown in the leftmost triangle. Points in resulting triangles represent individual nuclei remaining on the 300 × 300 grid at end of simulation. Note that there is significant overlap of points, thus the amount of black is not proportional to population size. Shading of pie charts shows percentage of nuclei with >66% mating fitness ($w_m$; region covered by bar). Shaded square indicates proportion of cells occupied at end of simulation. Simulations with no surviving population have no dots, and associated pie charts have a red cross.

## Standard dikaryon

When DMF is allowed through di-mon matings, the standard dikaryotic state found in nature, there is a strong drive for mating fitness. With polygenic fitness, the nuclei exclusively optimize mating fitness. However, if fitness variance is monogenic a clear polymorphism emerges with one set of nuclei strongly optimizing mating fitness, and the other optimizing spore production. Changing the dominance coefficient of the vegetative growth and spore production phenotypes had quantitative effects on the proportion of the nuclei optimizing mating fitness (*Figure 2* pie charts).

## Open dikaryon

When di-di matings are allowed, the hypothetical open dikaryon situation, the increased selection for mating fitness caused populations to collapse in most scenarios. In scenarios with low spore production (*Figure 2B*), the open dikaryon population crashes, regardless of dominance or genetics of fitness variance. When spore production is increased (*Figure 2A*), the populations survive as less spore production fitness is required, but all nuclei optimize mating fitness. When phenotypes are

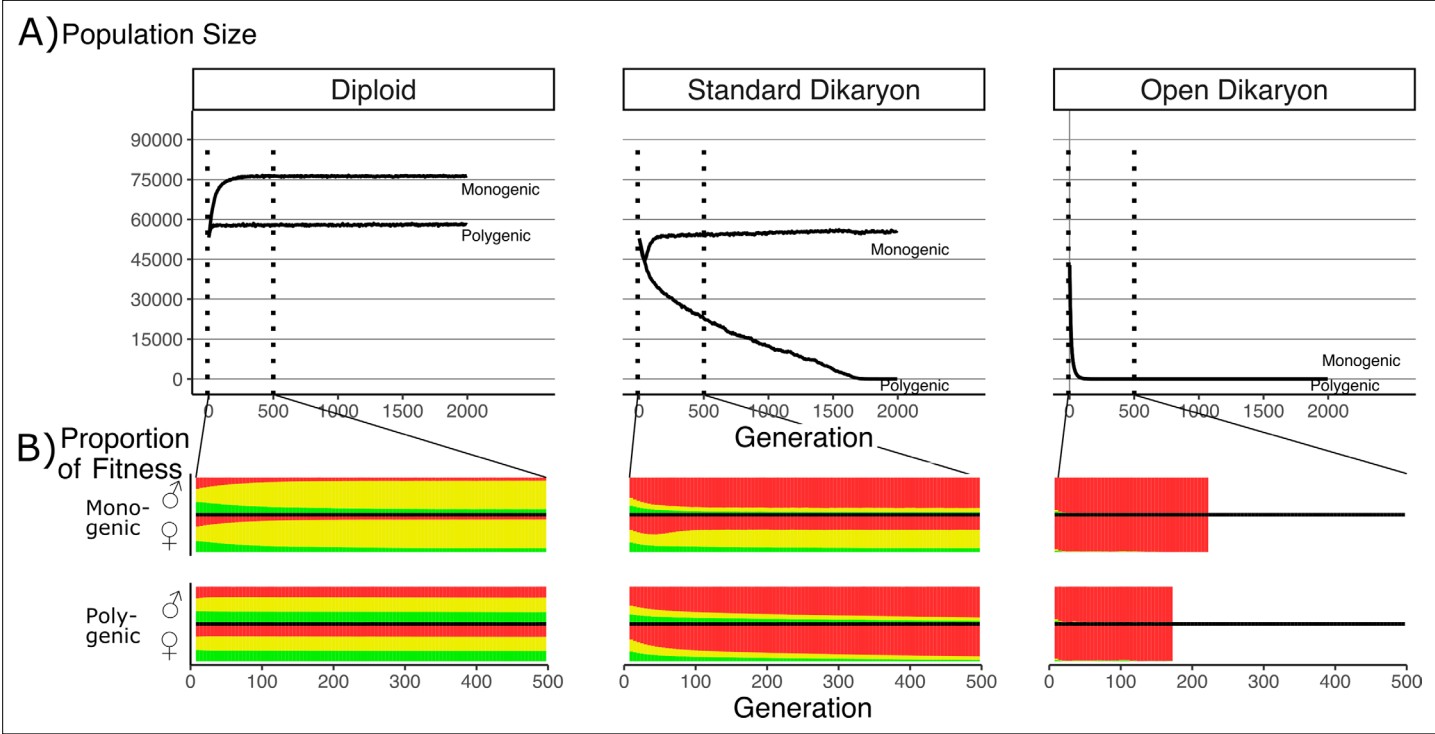

**Figure 3.** Effect of number of fitness loci on the impact of the fitness trade-off. (**A**) Population size with $\lambda$ varying from monogenic ($\lambda = 0$) to polygenic ($\lambda = 1$) across the three life strategies, initialized with 60,000 squares occupied. Dotted vertical lines indicate region shown in B. (**B**) Average proportion of fitness in nuclei per generation for the first 500 generations. Decreasing $\lambda$ values are shown on separate rows. The different fitness components are indicated by color: Red (top) is mating fitness, $w_m$, yellow (middle) is spore production, $w_r$; and green (bottom) is vegetative growth, $w_v$. White areas in open dikaryon are from dead populations. For every value of $\lambda$, the average fitness is calculated separately for nucleus 1 (the nucleus of the original monokaryon) and nucleus 2 (the fertilizing nucleus). Note that increased nonmating fitness in the open dikaryon with $\lambda = 0.99$ is due to extremely small population sizes.

recessive and fitness is monogenic, even this increased level of spore production is not sufficient to maintain the population.

To assess the stability of these results, we performed similar simulations with two further modifications to the global parameters, either increased basal growth rate, $g$ of 0.5 or increased mutational width, $\sigma$ of 0.05. Increased basal growth rate allowed the open dikaryon to maintain stable populations (*Figure 1—figure supplement 2*). Increased mutational width allowed the open dikaryon to persist under a wider set of parameters (*Figure 1—figure supplement 1*). For the diploid polygenic scenario, increased mutation width resulted in optimization of fitness around high spore production instead of a balance between the fitness components.

### Limited fitness loci partition mating fitness in the standard dikaryon

As the initial simulation showed clear differences between the monogenic and polygenic states, we more thoroughly investigated this influence. The population size dynamics are shown in *Figure 3A*, and average population fitness proportions are shown in *Figure 3B*. As population size results from the balance of dispersal/growth and the random death process, this size is an outcome of average fitness at the level of the mycelium (*Gilchrist et al., 2006*). Therefore, *Figure 3A* shows an element of average mycelial fitness, while *Figure 3B* shows the fitness components of the individual nuclei.

In the diploid situation, the monogenic scenario had an increased population size compared to the polygenic fitness scenario. *Figure 3B* shows that this increase in population size was co-incident with an increase in the average spore production fitness (yellow), and a roughly symmetrical decrease in vegetative growth and mating fitness. When simulated under similar parameters, the open dikaryon monogenic and polygenic situations resulted in population collapse, with rapid and irreversible increase in mating fitness (red) after only a few generations (*Figure 3B*).

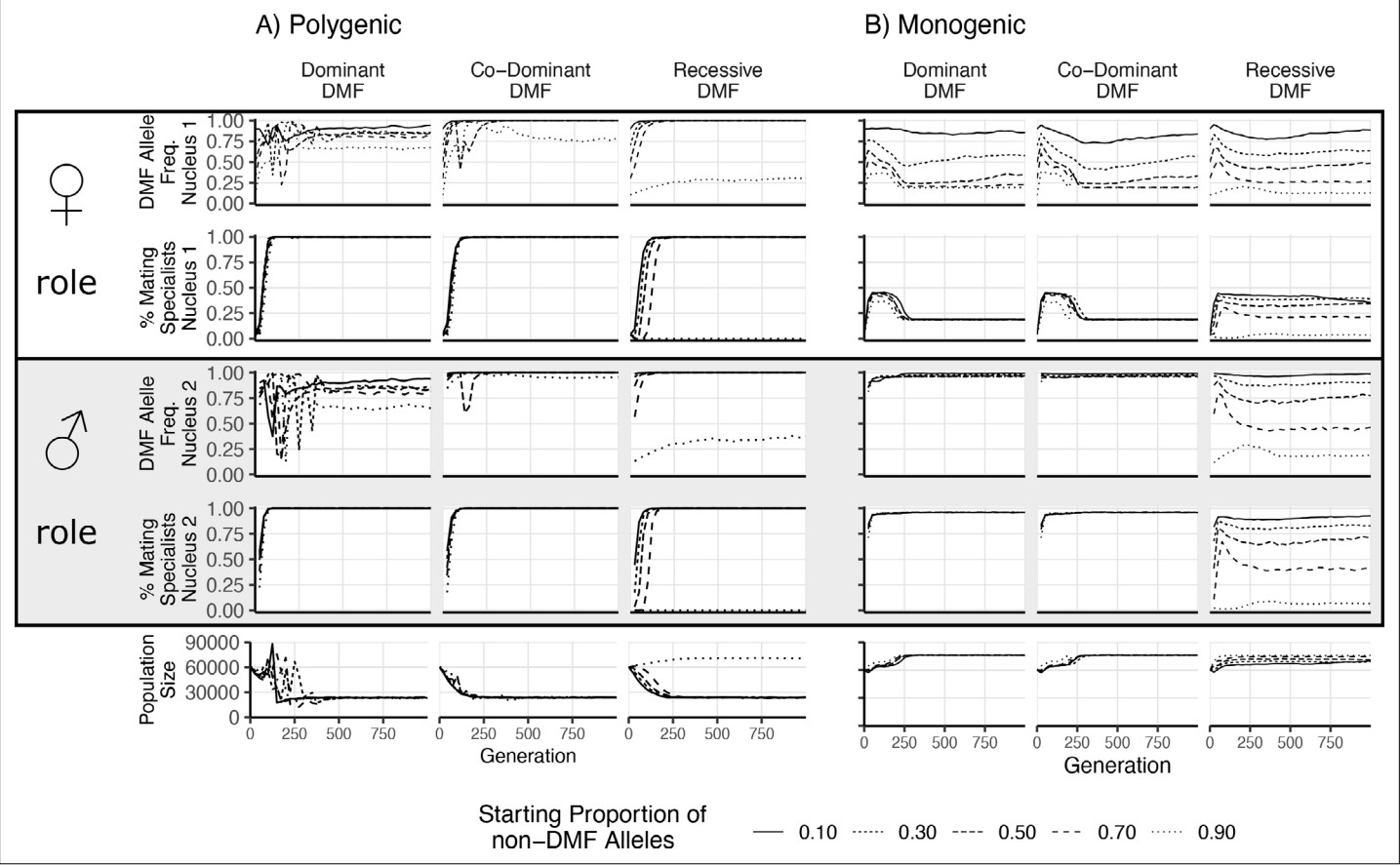

**Figure 4.** Results of direct competition between nuclei with and without dikaryotic male function (DMF) function. Different initial proportions of DMF and non-DMF nuclei are shown with different line styles. Initial DMF and non-DMF nuclei fitness parameters were drawn from the same distribution, with the same trade-off imposed. Proportion of DMF nuclei and proportion of parasitic nuclei (those with >66% mating fitness) are shown for the female nucleus (nucleus 1) in the top two rows, while the male nucleus (nucleus 2) is shown below. Note that since the population begins as completely homokaryotic, the nucleus 2 position is empty for generations 0. The bottom row shows the total population size.

The online version of this article includes the following figure supplement(s) for figure 4:

**Figure supplement 1.** Linearity of the fitness trade-off has limited effect on mating selection.

In the standard dikaryon, the polygenic situation leads to a continuous decline in population, as mating fitness increases in all nuclei (*Figure 3A*). The increased mating fitness is seen in both the male and female nuclei of the dikaryons in the population (*Figure 3B*). However, the monogenic situation allows for a polymorphism in the population between the fitness of the nuclei that were the original monokaryon (the female role) versus those of the fertilizing nucleus (the male role). The male role is then still performed by nuclei specialized for mating fitness, while the female role is performed by nuclei that retain a significant amount of spore production fitness. Interestingly, in the standard dikaryon the monogenic situation shows an initial increase in mating fitness in the female role as well, but this is purged after approximately 75 generations.

## Dikaryotic male function increases in frequency in direct competition

Although our simulations aimed to assess the balance between the levels of selection in the different life cycles, and not to study the transitions between them, it was important to assess whether the dikaryotic state was competitive under the imposed fitness trade-off. To this end, we competed mixtures with varying starting proportions of nuclei with and without DMF. We simulated DMF as a dominant, co-dominant (each nucleus acting independently), or recessive trait with the fitness trade-off as either a mono- or polygenic trait. As seen in *Figure 4*, in most cases when DMF is allowed, it increases in frequency (rows 1 and 3), but with different outcomes between the two nuclear positions. With a

polygenic fitness trade-off, the DMF allele frequency increases, particularly in the male nucleus. When the fitness trade-off is monogenic, the DMF allele still increases in the male nucleus, except when DMF is recessive, but the population size remains large. Interestingly, in this monogenic scenario, the initial increase in population size is delayed due to increasing mating fitness, in the female role to the detriment of reproductive or vegetative fitness components, in effect becoming parasitic nuclei. These nuclei are purged with the first 250 generations regardless of the starting proportion. This purging of high mating fitness genotypes in the female role is less effective in the recessive case, as deleterious alleles are hidden from selection at low frequencies. As the decreased population size with increased mating fitness is based on an assumed fitness trade-off, we tested the effects of a nonlinear trade-off (*Figure 4—figure supplement 1*). Regardless of the shape of trade-off surface, the DMF allele increased in frequency, and the effect of the nonlinearity only affected the rate of increase and the resulting population size.

## Effect of environmental pressures

To test the effect of environmental stability on the resulting populations, we varied the parameters of basal growth rate and the death rate, *g* and *d*, respectively. When these two rates are equal a given space has the same probability of dying as it does of being grown into again. When the death rate is higher than growth rate, then a larger proportion of cells will be available to be colonized by spores. As can be seen in *Figure 5A*, the populations were smaller with increasing death rate (y-axis) in all three types of life cycle, and low growth rate combined with high death rate led to population collapses (white squares in *Figure 5A*). The open dikaryon was susceptible to population crashes across a larger range of parameter values. In general, the diploid had the largest population size, the standard dikaryon intermediate, and the open dikaryon the smallest. Looking at the mating success, the proportion dikaryotic, in almost all scenarios the open dikaryon is 100% dikaryotic, while the diploid has a much higher percentage of monokaryons (*Figure 5B*). As it is difficult to visually compare the three scenarios based on color intensity, the overall averages of surviving populations are plotted in *Figure 5C*. This shows that indeed the open dikaryon has the highest proportion of dikaryons, but the lowest population size, and the opposite for the diploid. The standard dikaryon is intermediate between the other two scenarios. Notably, there seems to be a negative correlation between population size and proportion dikaryon across the three scenarios.

## Factors that reduce spread of genotypes detrimental to the dikaryon

We investigated potential factors that in the standard dikaryon may influence the proportion of nuclei maximizing their mating fitness, which is detrimental to the dikaryon due to the imposed fitness trade-off. We varied four factors: the number of mating types, total potential spore production, the basal growth rate, and the phenotypic dominance (*Figure 6*). As the number of loci affecting this fitness trade-off is apparently an important factor for the levels of mating fitness, we simulated across a range of $\lambda$ values, from $\lambda = 1$ the monogenic situation, to the polygenic scenario of $\lambda = 0$. The results showed little difference below $\lambda$ values of 0.75 (*Figure 6A*). With increasing $\lambda$, the phenotypes are more unevenly distributed among nuclei. The results in *Figure 6A* are resemble those from *Figure 3B*, except that *Figure 6A* shows different categories of percentage mating fitness per nucleus, while population averages for the three fitness components are shown in *Figure 3B*. From these linkage values, we selected $\lambda = 0.95$ which showed almost all nuclei having >90% mating fitness, $\lambda = 0.99$ which showed nuclei with a range of mating fitness, and $\lambda = 1.00$ where a smaller proportion of nuclei optimized mating fitness over 90% (*Figure 6B*). These scenarios resemble the single locus scenarios, but with offspring phenotypes that include a small amount of mixing between the parental phenotypes.

As individuals with the same mating type cannot mate, reduction of the number of mating types could be expected to reduce selection for mating fitness, as more spaces would be occupied by incompatible mates thus leaving fewer compatible mates for a given focal female. However, we saw little effect of this, and the only difference noticeable was a slight decrease of mating fitness with only two mating types when linkage was 0.99 or 1.00, but a slight increase in mating fitness when linkage was 0.95. Similarly, increasing spore production had a mild affect, with increased proportion of nuclei maximizing mating fitness when linkage was 0.95.

Increasing the growth rate also decreased the nuclei maximizing mating fitness generally, particularly with values of g greater than 0.5. Particularly with the highest linkage values this decrease

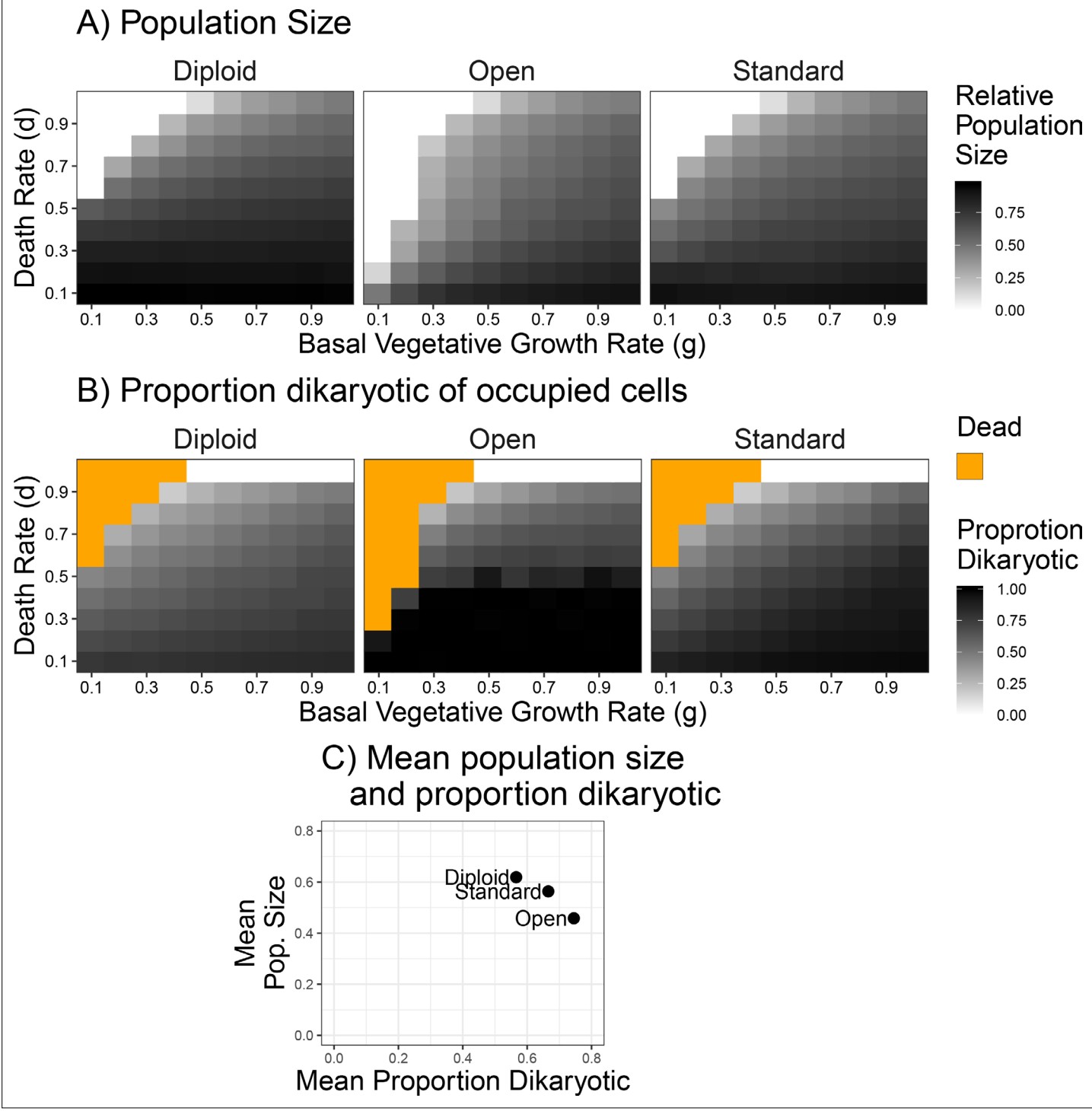

**Figure 5.** Population size and mating success with differing environmental influences. (**A**) Population size after 500 generations across a range of death and growth rates. Intensity of black is proportional to population size. (**B**) Mating success of simulations from panel (A), but intensity of black refers to proportion of occupied cells that are dikaryotic. Orange values indicate simulations had no surviving cells. (**C**) Grand mean of values used in (A) and (B), grouped by scenario to highlight the relative differences in mating success and population size between the three scenarios.

was more quantitative, with a similar number of nuclei being selected for mating fitness but instead resulting in less specialization. Increasing the dominance, especially above 0.5, resulted in increasing proportion of mating fitness specialists, particularly with higher linkage. To investigate the role of habitat fragmentation, we also experimented with splitting the grid into square patches of sizes from

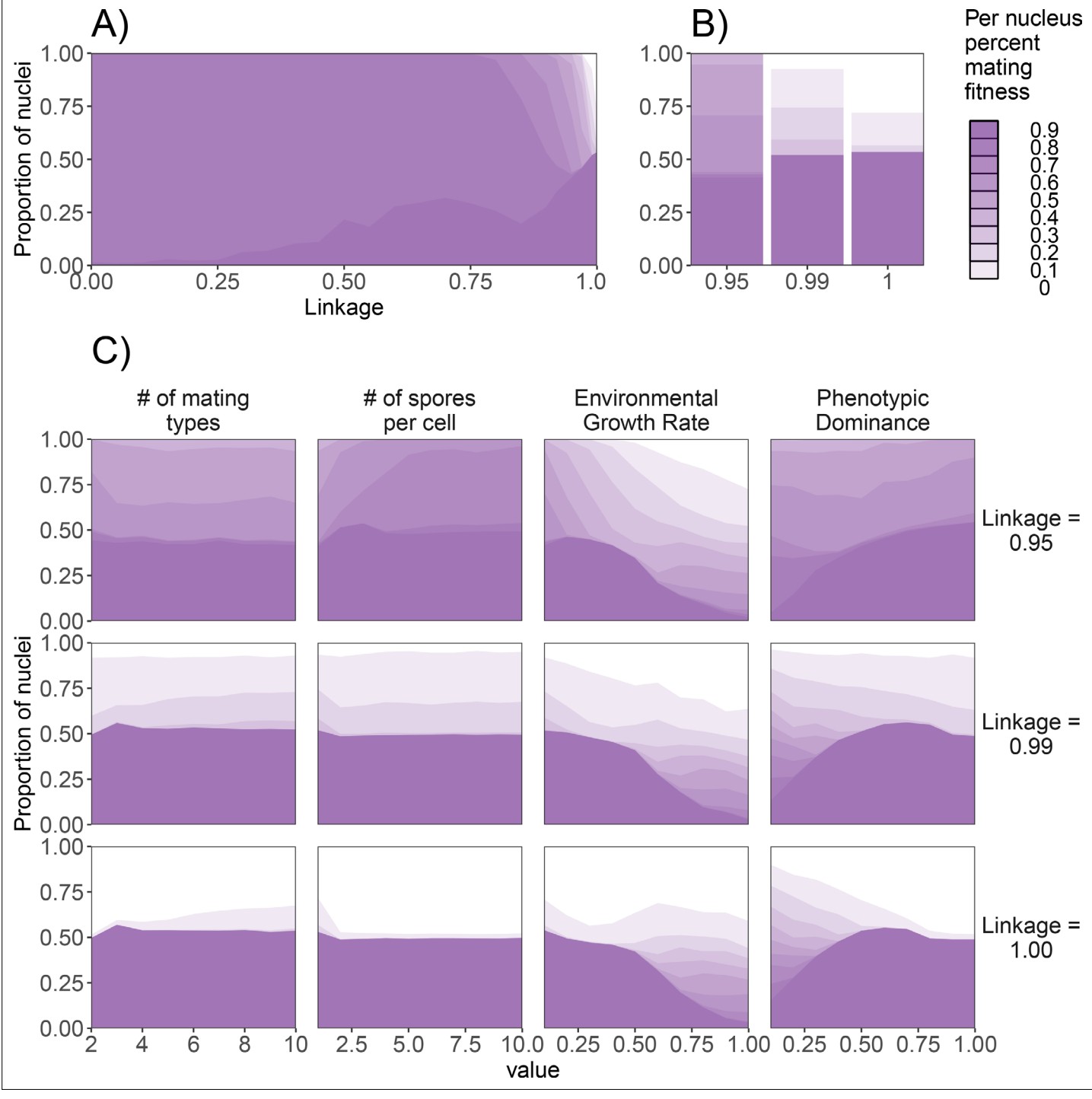

**Figure 6.** Effect of genetic factors on the prevalence of nuclei with detrimental levels of mating fitness. For all subpanels, mating fitness is binned from all nuclei remaining at the end of 1000 generations. (**A**) Effect of $\lambda$ on mating fitness in the standard dikaryon. (**B**) Mating fitness distributions for $\lambda$ values of 0.95, 0.99, and 1.00 which are selected to highlight differences in outcomes. (**C**) Effect of parameters on the resulting mating fitness in the population in the standard dikaryon scenario. Rows show results under linkage values of 0.95, 0.99, and 1.00. Columns from left to right show results of modifying the number of mating types, spore production, growth rate, and dominance. Shading is proportional to mating fitness, calculated in steps of 10% across four replicates. White area represents nuclei with <10% mating fitness.

2 × 2 up to 5 × 5, separated by strips of uninhabitable area (data not shown). Spores were capable of crossing this uninhabitable area, but mycelia could not grow across. This structured landscape resulted in increased variation between simulations, but we did not find qualitatively different results in the presence of nuclei maximizing mating fitness (data not shown).

## Discussion

The balance of evolutionary forces acting upon this unique dikaryotic life history has been unclear. Here we assess the impact of different levels of selection, nuclear, and mycelial, on the dikaryotic life cycle. The benefits of increased mating opportunities for a nucleus in a dikaryon seem obvious but the potential costs are perhaps less clear. Importantly, the dikaryotic state results in an increased proportion of individuals capable of sexual reproduction, particularly the male role, in the population compared to a diploid, due to the increased mating opportunities. As monokaryons generally have no reproductive output in Basidiomycetes, the resources they control (e.g. a section of a tree stump, or ectomycorrhizal tree roots) represent an opportunity for nuclei from dikaryotic mycelial neighbors. Since the nuclei within a dikaryon have low genetic relatedness, competition between them for these additional mating partners provides a tension, potentially resulting in genomic conflict. Our results show that even in the face of an imposed fitness trade-off at the organismal level, a dikaryotic life cycle will still select for mating success. Our results further show that the environment influences the costs and benefits of this dikaryotic life cycle as the density of individuals influences both the importance of mycelial level fitness and the chances for repeated matings. Our experiments with splitting the grid into patches indicates that our results are robust to some degree of spatial heterogeneity. However, extreme distribution patterns of suitable growth conditions could lead to genetic differentiation between patches, favoring selection between patches for mycelial fitness components.

A clear result of our simulations is the strong drive for mating fitness in the standard and open dikaryon. Under all tested scenarios and parameters, the selection for mating fitness resulted in most nuclei specializing in mating fitness. Depending on the parameters, this even resulted in population collapse. The selection benefit of di-mon mating was particularly visible when individual nuclei with and without DMF are competed directly. The DMF allele spreads through the population, even though it comes with the trade-off, regardless of the shape. This is because the adjacent monokaryons provide a reliable route for gene copy increase of the genotypes with the DMF allele. The marginal decrease in fitness from vegetative growth/spore production, due to the imposed trade-off, is outweighed by the benefit of additional matings. This drive for mating fitness is even stronger in the open dikaryon, leading to population collapse under a wider range of parameter settings than the standard dikaryon. It is important to note that the diploid scenario also selects for increased mating fitness, although with less strength, as matings between monokaryons can also include multiple partners with differing mating fitness, which has been shown in yeast to strongly select for mating pheromone production (*Rogers and Greig, 2009*). The fitness decrease at the dikaryotic level was dependant on the dominant/recessive nature of fitness components. While we did not test overdominant scenarios, the specialization of genomes leading to vigorous hybrid dikaryons, we do not expect it would qualitatively change the results, but would lead to increased mating fitness as even less residual vegetative or spore production fitness would be required for sustained population growth.

Our simulations also indicate that the nature of fitness inheritance has a strong effect on any deleterious effects of nuclear selection. When offspring are similar to parents due to linkage of fitness components, intermediate (i.e. nonparental) offspring are restricted to those produced by mutation. The meiotic progeny is differentiated into two types, producing either competitive or noncompetitive gametes. However, when fitness components are unlinked across multiple loci, matings result in gametes with competitive alleles at on average half of the loci segregating between the parents, resulting in very low phenotypic variance among offspring, resulting in uniform specialization for mating fitness. While our simulations used a blending inheritance model, this has been shown not to have a large effect on resulting social interactions when compared to biologically realistic Mendelian particulate inheritance (*Gardner, 2011*). The fact that our simulations compared fitnesses of parents to determine the fitness of offspring, in the absence of a genetic model, makes this work similar to the 'phenotypic gambit' (*Grafen, 1984*; *Grafen, 2014*). Although future work would likely benefit from a more explicit genetic determinism, such phenotypic gambits are often useful as a first approximation to clarify future hypotheses (*Grafen, 2014*). Of potential interest, backcrossing of wild isolates has

shown that variation in mating success potentially resides near/within the mating-type locus (*Nogami et al., 2002*). In general, Basidiomycete mating-type loci are idiomorphic (highly polymorphic alleles) and have suppressed recombination, spanning from a general 50–150 Kb up to several Mb (*Branco et al., 2017*; *Brown and Casselton, 2001*). Future studies should investigate if alleles conferring mating success truly reside inside or near the mating locus, and if so, this is correlated with growth and/or reproduction. If increased mating fitness was negatively correlated with other growth characteristics, similar to the simulations here, such 'parasitic' mating-types alleles could be stable in a population since the trade-offs between reduced growth and increased mating success would be linked within a single nonrecombining locus.

A key assumption in our model is the trade-off between fitness components. While we do not specify the underlying nature of the trade-off, our assumption is consistent with antagonistic pleiotropy of the genes involved (i.e. an allele can be optimized for either fitness component but not all). This may seem a strong assumption. However, even without antagonistic pleiotropy, trade-offs are likely (*Garland, 2014*; *Matthewson and Weisberg, 2008*; *Stearns, 1989*) particularly if selection occurs at multiple levels (*Booth, 2022*; *Lewontin, 1970*). Increased selection at the level of the nucleus may also allow accumulation of mutations deleterious for dikaryon fitness, resulting in a trade-off. An experimental evolution study attempting to select for competitive nuclei failed to find a trade-off with mycelium-level fitness components (*Nieuwenhuis et al., 2018*). However, in this experiment simultaneous selection for multiple fitness components could not be excluded. Supporting the presence of a cost for nuclear selection, modifying our trade-off with nonlinearity resulted in population decreases proportional to the nonlinearity. This shows that the shape of the fitness trade-off has a quantitative, but not qualitative effect on selection for mating success. Importantly, nuclei capable of DMF are selected for, even when the fitness trade-off is strongly nonlinear, leading to a strong cost of mating fitness. In a population of diploid individuals, an allele conferring DMF has increased fitness despite the associated trade-offs.

The relationship between the different fitness components of dikaryons requires further study. Our model crucially assumes that the competitive success of nuclei during di-mon matings is the result of competition between the two nuclei in the dikaryon. Alternatively, the interaction between a nuclear type and the receiving monokaryon may affect the outcome of nuclear competition, effectively a form of female choice by the receiving monokaryon (*James, 2015*; *Kües, 2015*; *Nieuwenhuis et al., 2011*; *Nieuwenhuis and Aanen, 2012*). The B locus, one of the two loci determining the mating type of tetrapolar species, encodes both pheromones and pheromone receptors. While a single pheromone-receptor interaction is sufficient for sexual compatibility, typically there is high redundancy in the number of compatible pheromone-receptor interactions, with an excess of compatible pheromones (*Kües, 2015*; *Nieuwenhuis and Aanen, 2012*). It has been argued that this redundancy is a consequence of female choice, and that pheromones may function as an honest signal of nuclear quality (*James, 2015*; *Rogers and Greig, 2009*). If female choice is important, a trade-off between nuclear- and mycelial-level fitness components may not be expected, but instead a positive relationship due to the 'good genes' model (*Zahavi, 1975*). Under this model of female choice, nuclear competition could instead have positive effects on mycelial fitness. However, the exact consequences of different assumptions on the relationship between fitness components should be modeled.

While it is often assumed that in fungi the mycelial individual (whether monokaryon or dikaryon) is the unit of selection, in multinucleate fungi this is not the only level (*Booth, 2022*; *Lewontin, 1970*). Our model shows an example of how selection at the level of the nucleus, from competition between unrelated nuclei for mating, occurs despite being detrimental to the level of the dikaryon. Nuclear selection has been shown to act rapidly in hyphal fungi, where cheater nuclei can be selected in experimental evolution (*Bastiaans et al., 2016*), but also in natural isolates of a dikaryotic ascomycete, *Neurospora tetrasperma* (*Meunier et al., 2018*). An important factor in dikaryotic fitness considerations is the level of phenotypic dominance, which can mask deleterious mutations. The prevalence of dominant phenotypes in Basidiomycetes is currently unknown, although recent work has found little support for dominant-recessive relationships (*Clergeot et al., 2019*; *Hiscox et al., 2010*; *Nobre et al., 2014*).

The idea of nuclear selection was not initially connected with fitness trade-offs, instead predicting ever increasing sexual fitness as a result of internuclear competition (*Raper, 1966*). A recent reappraisal of internuclear competition phrased it as the 'selfish nucleus model', imagining a scenario

where mating fitness could select for nuclei detrimental to the dikaryon, similar to our simulation design (*James et al., 2009*). Our results, with open dikaryons resulting in detrimental levels of nuclear selection, are consistent with the intuition of *James et al., 2009*, who postulated the occurrence matings between dikaryons '*provided the negative effects of nuclear competition are not also increased*' (*James et al., 2009*). Our simulations show that if such negative effects of nuclear competition exist, then open dikaryons will suffer from reduced fitness. Therefore, it is crucial to establish the relationship between nuclear competitiveness in di-mon interactions, and the fitness of the resulting dikaryon. As mentioned above, an alternative model is that the interaction between fertilizing nuclei and receiving monokaryon determines nuclear competitive success, which would be a form of female choice based on gamete testing.

Nuclear exchange between dikaryons, di-di matings in our simulations, has been demonstrated in natural settings (*Hansen et al., 1993*; *Johannesson and Stenlid, 2004*). However, we would argue this is not equivalent to an open dikaryon. These examples of nuclear exchange between dikaryons are documented in *Heterobasidion* species, a common Basidiomycete study system, but one in which dikaryons develop abundant monokaryotic hyphal sectors (*Johannesson and Stenlid, 2004*). Presumably the observed nuclear exchange between dikaryons occurs through monokaryotic hyphal intermediates. This inclusion of monokaryotic hyphal intermediates makes this more similar to either di-mon or mon-mon mating, depending on how the monokaryotic sections are fertilized. Crucially, there is a qualitative difference between di-di and di-mon matings through monokaryotic intermediates, as novel dikaryons from the latter will be somatically incompatible with the two established dikaryotic progenitors, having no spatial access to further resources (*Johannesson and Stenlid, 2004*). Such segregation into monokaryotic hyphae with subsequent mating would be compatible with the model recently suggested as 'somatic hybridization' in rust fungi (*Li et al., 2019*; *Nelson et al., 1955*; *Wu et al., 2019*). These apparent di-di matings may not experience the same evolutionary forces as if true di-di matings were allowed, due to the physical restriction of the novel dikaryons.

Our simulations aimed to understand the potential consequences of nuclear selection in mushroom-forming Basidiomycetes but may also have relevance for other fungal groups. The widespread plant pathogenic rust fungi also have a dikaryotic stage. Nuclear exchange between these rust dikaryons is also reported, 'somatic hybridization', in the production of novel pathogenic rust races (*Burdon et al., 1981*; *Li et al., 2019*; *Nelson et al., 1955*; *Wu et al., 2019*). It is difficult to reconcile the occurrence of somatic hybridization with our results showing that open dikaryons are highly susceptible to nuclear selection. One possibility is that somatic hybridization is exceedingly rare, as a single somatic hybridization event could produce a rust race able to grow on otherwise resistant hosts. Another possibility would be monokaryotic intermediate hyphae, similar to that in *Heterobasidion* (*Hansen et al., 1993*; *Johannesson and Stenlid, 2004*). Also, recent evidence indicates that the abundant Arbuscular Mycorrhizal fungi from the phylum Mucoromycota also exist in a dikaryotic state (*Ropars et al., 2016*). While hyphal fusion occurs in this lineage (*Croll et al., 2009*), the potential for di-mon, or even di-di matings, remains unknown.

The open dikaryon of our model corresponds to the 'unit mycelium' concept formulated to explain observations of cell fusions in cultured fungal material (*Buller, 1934*). This 'unit mycelium' concept was the working paradigm over many decades, with its assumption of cooperation between genetically unrelated individuals. However, accumulating evidence from natural isolates, showing that neighboring dikaryotic mycelia were instead distinct, led to the influential synthesis of the 'individualistic mycelium' (*Rayner, 1991*). This model of distinct individuals separated by allorecognition responses was consistent with the observation of somatic incompatibility between dikaryons, with fusions between nonself individuals leading to cell death, not cooperation (*Booth, 2022*). The 'individualistic mycelium' model is not only consistent with experimental evidence but also compatible with predictions that cooperation between nonrelated dikaryons to be exploited almost immediately (*Czárán et al., 2014*).

This persistent dikaryotic state has been retained for over 400 million years (*Chang et al., 2015*), yet few extant Basidiomycete species are diploid (*Anderson and Kohn, 2007*). The retention of the dikaryotic state in the vast majority of Basidiomycetes leaves open the potential for mycelium-level fitness costs of nuclear selection due to di-mon matings. While there is little data on the prevalence of di-mon matings in the field, they may be more common than is generally assumed (*Nieuwenhuis et al., 2013*). Given the potential for strong selection for mating fitness within a dikaryon, it is crucial to determine the relationship between nuclear competitiveness and mycelium-level fitness.

## Conclusion

Potential benefits of a dikaryon stage have been a subject of discussion for some time. The presence of the dikaryotic state emphasizes the level of selection on the individual nuclei, something that is often overlooked in evolutionary discussions of fungi. The persistent association between unrelated haploid nuclei will invariably select for mating success, potentially to the detriment of the individual. Our results show the potential costs of selection at the level of nuclei for mating for mycelium-level fitness components, and how those can be reduced. First, restricting fertilization by dikaryons to monokaryons reduces the level of nuclear mating fitness and maintains a higher mycelium-level fitness, compared to completely free exchange between dikaryons. Second, if the variance in mating fitness and its associated trade-off is restricted to a single locus, the costs to mycelial level fitness are reduced. Most importantly, establishing the relationship between the different fitness components is crucial to understand the potential consequences of nuclear competition in the dikaryotic life cycle. We also hypothesize that recent examples of dikaryon nuclear exchange are due to monokaryotic intermediates or result from extremely rare events. Our results show that the consequences of competition between the unrelated haploid nuclei in a dikaryon can be severe, and there must exist mechanisms to police it.

## Data availability

Source to run simulations, as well as scripts to produce figures and analysis are found at https://github.com/BenAuxier/Basid.Sex.Sim (*Auxier, 2021a*).

## Acknowledgements

We gratefully acknowledge the input from anonymous reviewers of a previous version of this manuscript whose critical comments led to substantial improvements in content and clarity. Support to TC from grant # K140901 (K124438) funded by NKFIH, Budapest, Hungary, is acknowledged. Support to BA from NWO (ALGR.2017.010) and to DKA from NWO (VICI; NWO86514007) is also acknowledged.

## Additional information

### Funding

| Funder | Grant reference number | Author |
|---|---|---|
| National Research Development and Innovation Office | K140164 | Tamás L Czárán |
| Nederlandse Organisatie voor Wetenschappelijk Onderzoek | ALGR.2017.010 | Benjamin Auxier Benjamin Auxier |
| Nederlandse Organisatie voor Wetenschappelijk Onderzoek | NWO86514007 | Duur K Aanen |

The funders had no role in study design, data collection and interpretation, or the decision to submit the work for publication.

### Author contributions

Benjamin Auxier, Data curation, Formal analysis, Investigation, Methodology, Software, Visualization, Writing – original draft, Writing – review and editing; Tamás L Czárán, Conceptualization, Data curation, Formal analysis, Investigation, Methodology, Resources, Software, Writing – review and editing; Duur K Aanen, Conceptualization, Investigation, Project administration, Supervision, Writing – review and editing

### Author ORCIDs

Benjamin Auxier  http://orcid.org/0000-0002-7743-0610
Tamás L Czárán  http://orcid.org/0000-0002-2722-6208

Duur K Aanen http://orcid.org/0000-0002-5702-1617

**Decision letter and Author response**
Decision letter https://doi.org/10.7554/eLife.75917.sa1
Author response https://doi.org/10.7554/eLife.75917.sa2

## Additional files

### Supplementary files
• MDAR checklist
• Source code 1. Pseudocode for the simulation algorithm.

### Data availability

The current manuscript is a computational study, so no data have been generated. Simulation code for performing simulations as well as scripts to produce figures and analyses are available in Github repository https://github.com/BenAuxier/Basid.Sex.Sim (copy archived at swh:1:rev:e407cb4cf2f4d63ff41e5caf1e9a90a6a79131f0).

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
