## [Editor Report]

Unions between equal partners can be destabilized by matings with third parties. In this paper the authors demonstrated that in fungi, 'stable unions' of two nuclei (dikaryons) are predicted to experience costs to vegetative fitness from investment in such mating opportunities. 'Open unions', in which third parties have access to the resources of established partnerships, are evolutionarily highly unstable. This paper will be of general interest to those who study evolutionary conflicts and to fungal geneticists.

---

## [Decision Letter]

**Decision letter after peer review:**

Thank you for submitting your article "Living apart together: modelling the consequences of the dikaryotic life cycle of mushroom-forming fungi for genomic conflict" for consideration by *eLife*. Your article has been reviewed by 2 peer reviewers, and the evaluation has been overseen by a Reviewing Editor and Aleksandra Walczak as the Senior Editor. The following individuals involved in review of your submission have agreed to reveal their identity: Timothy Y. James (Reviewer #1); David Haig (Reviewer #2).

Essential revisions:

1) Clarify your descriptions on fitness (see Reviewer #1's public review point 1),

2) Clarify the role of spatial heterogeneity in your model (Reviewer #1's public review point 2), and

3) Consider the scenario of overdominance (raised by Reviewer #2).

*Reviewer #2 (Recommendations for the authors):*

This was a 'blast from the past'. I went to England in 1990 intending to work on basidiomycete mating systems, including the Buller phenomenon, but genomic imprinting in mammals rapidly consumed my time and I never seriously took up the project. All that survives of that project is a brief discussion of the production of asexual spores by dikaryons in Journal of Theoretical Biology 153: 542. Asexual reproduction of conidia is another another form of 'reproduction on the side' in which there could be conflict between the partners in a dikaryon. For this reason, I suspect, asexual propagules are frequently produced by monokaryons but rarely by dikaryons.

The dikaryotic vegetative fitness is modeled as the average of the monokaryotic vegetative fitnesses plus a dominance term. It might be interesting to consider overdominance because a possible advantage of forming a dikaryon is complementation of weaknesses of each of the dikaryons.

'ditributions' at line 182.

line 183: mutation step does not appear to be shown in Figure 1A. This possibly refers to Figure 2A.

---

## [Author Response]

Essential Revisions (for the authors):1) Clarify your descriptions on fitness (see Reviewer #1's public review point 1),

We have updated the fitness descriptions, as well as an updated description of the λ parameter, explained more fully in response to Reviewer #1’s point 1.

2) Clarify the role of spatial heterogeneity in your model (Reviewer #1's public review point 2), and

We have added some discussion of preliminary experiments regarding splitting our grid into patches, which did not result in substantive changes. We have added this topic to the Discussion as well. We have also emphasized that as our spore production is calculated per cell, that larger organisms have increased reproductive output.

3) Consider the scenario of overdominance (raised by Reviewer #2).

We have added discussion about this topic to the second paragraph of the Discussion, but we felt simulations regarding this would likely fall outside the scope of this manuscript. Using the phenotypic gambit here as we do complicates implementations of overdominance.

Reviewer #2 (Recommendations for the authors):This was a 'blast from the past'. I went to England in 1990 intending to work on basidiomycete mating systems, including the Buller phenomenon, but genomic imprinting in mammals rapidly consumed my time and I never seriously took up the project. All that survives of that project is a brief discussion of the production of asexual spores by dikaryons in Journal of Theoretical Biology 153: 542. Asexual reproduction of conidia is another another form of 'reproduction on the side' in which there could be conflict between the partners in a dikaryon. For this reason, I suspect, asexual propagules are frequently produced by monokaryons but rarely by dikaryons.The dikaryotic vegetative fitness is modeled as the average of the monokaryotic vegetative fitnesses plus a dominance term. It might be interesting to consider overdominance because a possible advantage of forming a dikaryon is complementation of weaknesses of each of the dikaryons.

We have added a section in the discussion regarding our expectations regarding the influence of overdominance. We expect this would only allow further increases in mating fitness, and would not be a qualitative difference from the results shown here.

'ditributions' at line 182.

Fixed

line 183: mutation step does not appear to be shown in Figure 1A. This possibly refers to Figure 2A.

An old version of this figure was uploaded in error, the correct version is inserted now.